# Comparative Colorimetric Sensor Based on Bi-Phase γ-/α-Fe_2_O_3_ and γ-/α-Fe_2_O_3_/ZnO Nanoparticles for Lactate Detection

**DOI:** 10.3390/bios12111025

**Published:** 2022-11-16

**Authors:** Ricardo A. Escalona-Villalpando, Karen Viveros-Palma, Fabiola I. Espinosa-Lagunes, José A. Rodríguez-Morales, Luis G. Arriaga, Florika C. Macazo, Shelley D. Minteer, Janet Ledesma-García

**Affiliations:** 1División de Investigación y Posgrado, Facultad de Ingeniería, Universidad Autónoma de Querétaro, Santiago de Querétaro 76010, Mexico; 2Centro de Investigación y Desarrollo Tecnológico en Electroquímica, Pedro Escobedo 76703, Mexico; 3Department of Chemistry, University of Utah, 315 South 1400 East, Salt Lake City, UT 84112, USA

**Keywords:** lactate colorimetric sensor, bi-phase γ-/α-Fe_2_O_3_, peroxidase-like activity, lactate oxidation, bi-phase γ-/α-Fe_2_O_3_/ZnO

## Abstract

This work reports on Fe_2_O_3_ and ZnO materials for lactate quantification. In the synthesis, the bi-phase γ-/α-Fe_2_O_3_ and γ-/α-Fe_2_O_3_/ZnO nanoparticles (NPs) were obtained for their application in a lactate colorimetric sensor. The crystalline phases of the NPs were analyzed by XRD and XPS techniques. S/TEM images showed spheres with an 18 nm average and a needle length from 125 to 330 nm and 18 nm in diameter. The γ-/α-Fe_2_O_3_ and γ-/α-Fe_2_O_3_/ZnO were used to evaluate the catalytic activity of peroxidase with the substrate 3,3,5,5-tetramethylbenzidine (TMB), obtaining a linear range of 50 to 1000 μM for both NPs, and a 4.3 μM and 9.4 μM limit of detection (LOD), respectively. Moreover, γ-/α-Fe_2_O_3_ and γ-/α-Fe_2_O_3_/ZnO/lactate oxidase with TMB assays in the presence of lactate showed a linear range of 50 to 1000 µM, and both NPs proved to be highly selective in the presence of interferents. Finally, a sample of human serum was also tested, and the results were compared with a commercial lactometer. The use of ZnO with Fe_2_O_3_ achieved a greater response toward lactate oxidation reaction, and has implementation in a lactate colorimetric sensor using materials that are economically accessible and easy to synthesize.

## 1. Introduction

Lactate is a metabolite resulting from the anaerobic process of the respiratory chain in cells. Its quantifying is related to clinical diagnosis, the study of metabolic diseases, and continuous monitoring in sports medicine, trauma, and the food industry [1]. Lactate concentration is usually determined by HPLC, fluorometry, colorimetric tests [2], chemiluminescence, magnetic resonance spectroscopy, or electrochemical methods [3]. Each technique has certain advantages; although in some cases, the equipment is complex and expensive. In the case of colorimetric techniques, they are among the most required, because they are simpler and the equipment is economically accessible, but sometimes, its low sensitivity, quantification ranges, and limits of detection are parameters that are in constant investigation and improvement [4,5]. Conventionally, lactate has been quantified in blood or serum at concentrations of 0.5–2.5 mM, but this technique has the drawback that it is invasive to obtain the sample. Therefore, the trend in colorimetric quantification of lactate has been focused on sweat with concentration ranges from 15 to over 65 mM [6], although not all already reported sensors reach these quantifiable values [7]. Another of the biological fluids where the lactate could be measurable is the saliva, where the concentration varies between 0.11 to 0.56 mM, so it can be challenging to quantify it by colorimetric methods [8].

For the colorimetric quantification of lactate, the lactate oxidase enzyme (LOx) is usually used, where pyruvate and hydrogen peroxide (H_2_O_2_) result as secondary products. From this reaction, H_2_O_2_ is used as a substrate to react with a metal or metallic oxide with peroxidase-like activity and a chromogenic substance such as TMB that can be measured by UV-Vis spectroscopy [4]. Peroxidase-like iron oxides or iron-modified oxides have preferably been used due to their catalytic activity that depends on their intrinsic properties. These materials are also economically more viable compared to precious metals such as gold [9,10] or silver [11,12]. The peroxidase-like activity of Fe_3_O_4_ followed a trend of spheres > triangular plates > octahedral [13], with a smaller size to 30 nm [14,15] that has been published. Fe_2_O_3_ has also been reported from alpha and gamma phases, although with a low peroxidase-like activity for this last phase [16,17]. Binding with another material such as CeO_2_, CuO, Co_3_O_4_, V_2_O_5_, and ZnFe_2_O_4_ [18] can increase the catalytic activity. In addition, using other types of oxides, such as SiO_2_ or ZnO, facilitates a union with proteins improving mechanical and photovoltaic properties [19,20].

In this work, the synthesis of Fe_2_O_3_ and Fe_2_O_3_/ZnO by a simple and controlled precipitation method that could obtain the phases γ-/α-Fe_2_O_3_ and γ-/α-Fe_2_O_3_/ZnO is presented. Both materials were evaluated as peroxidase mimics through the catalytic oxidation of hydrogen peroxide. TMB concentration, wavelength, and H_2_O_2_ concentrations were standardized. Subsequently, both NPs were used to quantify lactate with the LOx enzyme (Figure 1), which has been poorly studied compared to other molecules such as glucose [21]. Therefore, we report the modification of Fe_2_O_3_ with ZnO in order to improve the stability of the enzyme and catalytic activity through the peroxidase activity of Fe_2_O_3_ used as a colorimetric lactate biosensor.

## 2. Materials and Methods

### 2.1. Materials

All used reagents are analytical grade from commercial sources. FeCl_3_·6H_2_O, TMB, H_2_O_2_ (30%), NaOH, LOx from Aeroccoccus viridans, Zn(O_2_CH_3_)_2_·2H_2_O, anhydrous sodium acetate, acetic acid, 3,3,5,5-tetramethylbenzidine TMB, and dimethyl sulfoxide (DMSO) were obtained from Sigma-Aldrich. Phosphate buffer solution (PBS) was prepared using Na_2_HPO_4_ and KH_2_PO_4_ (pH 7.4) from J.T. Baker. Acetate buffer solution (HAc-NaAc) was prepared from acetic acid and sodium acetate (pH 5) obtained from J. T. Baker. All aqueous solutions were prepared using deionized water DI (σ ≥ 18 MΩ cm).

### 2.2. Synthesis of Fe_2_O_3_ and Fe_2_O_3_/ZnO

Briefly, Fe_2_O_3_ preparation consisted of Fe_2_Cl_3_·6H_2_O salt (1 mol) in DI water (10 mL) in an ultrasonic bath for 30 min, keeping the temperature below 25 °C. Subsequently, 1.5 M NaOH (25 mL) was added slowly using a programmed injection pump for 1 h. Finally, the solution was maintained at 50 °C for 3 h under constant stirring. The precipitate was collected and washed several times with DI. The product was dried in an oven at 100° C for 24 h and then calcined at 500 °C during 4 h. For Fe_2_O_3_/ZnO nanoparticles, the synthesis procedure is as follows: 3 M NaOH (10 mL) was added by drop for 30 min using an injection pump to 1 mol Zn(O_2_CH_3_)_2_·2H_2_O in DI (10 mL) under constant sonication. At the same time, 1 mol Fe_2_Cl_3_·6H_2_O dissolved in 10 mL of DI and 10 mL 3 M NaOH was added by drop to the previous solution (injection pump) over a period of 30 min. Finally, the mixture was heated at 50 °C for 3 h. After that time, the precipitate was filtered and dried at 100 °C for 24 h and then calcined at 500 °C for 4 h in ramps with a temperature increase of 50 °C every 30 min.

### 2.3. Peroxidase-like Activity Measurements

The peroxidase-like behavior of the γ-/α-Fe_2_O_3_ and γ-/α-Fe_2_O_3_/ZnO NPs was analyzed spectrophotometrically at 653 nm through the H_2_O_2_ oxidation in the presence of TMB. The catalytic activity of γ-/α-Fe_2_O_3_ and γ-/α-Fe_2_O_3_/ZnO NPs toward TMB oxidation was studied using 5 mg mL^−1^ of NP’s (20 μL), 500 μM TMB in DMSO (30 μL), NaHOAc buffer (0.01 M, pH 4.0, 1.9 mL), and 30% H_2_O_2_ (30 μL). The wavelength was selected from a measurement sweep between 400 to 800 nm in the presence of 500 μM H_2_O_2_ using a UV-Vis spectrophotometer. The concentrations of oxidized TMB were quantified by UV–vis absorption using ε = 3.9 × 10^4^ M^−1^ cm^−1^ as the molar extinction coefficient [19]. The kinetic analysis using TMB as the substrate was performed varying the concentrations in the absence of TMB until 1500 μM. Similarly, the kinetics analysis with H_2_O_2_ was performed by varying the concentrations (0, 1, 5, 10, 50, 100, 250, 500, 750, 1000, and 1500 μM) for 10 min. The Michaelis–Menten constant was calculated using Lineweaver–Burk plots, 1/ν = Km/Vm (1/[S] + 1/Km); ν is the initial velocity, Vm is the maximal reaction velocity, [S] is the concentration of the substrate, and Km is the Michaelis constant. The determination of each sample was repeated three times.

### 2.4. Colorimetric Measurement of Lactate Using γ-/α-Fe_2_O_3_ and γ-/α-Fe_2_O_3_/ZnO NPs

Lactate detection was performed using 40 μL of LOx (100 U mL^−1^) in phosphate buffer solution (0.01 M PBS pH 7) using 200 μL of different lactate concentrations (0, 5, 10, 50, 100, 300, 500, 800, 1000, 3000, and 5000 μM). The solutions were incubated at 37 °C for 30 min. Later, 30 μL of TMB (500 μM), 20 μL of γ-/α-Fe_2_O_3_ and γ-/α-Fe_2_O_3_/ZnO (5 mg mL^−1^) NPs and acetate buffer (pH 4) were added into the LOx-lactate solution (final volume: 2 mL) and the solution was incubated at 37 °C for 15 min. Finally, the resulting solution was separated from the NPs by decanting, and then the absorbance was measured at 653 nm. The limit of detection (LOD) was calculated considering 3 σ/S; σ is the standard deviation of the blank signal and S the sensitivity obtained from the slope of the calibration cure, whereas the limit of quantification (LOQ) was calculated from 10 σ/S. This same calculation was used for peroxidase-like activity measurements.

### 2.5. Colorimetric Evaluation of Lactate Using γ-/α-Fe_2_O_3_ and γ-/α-Fe_2_O_3_/ZnO in the Presence of Interferents and Real Samples

To evaluate the selectivity of the colorimetric lactate sensor, 1 mM lactate was used in the presence of 0.5 mM interferents separately as dopamine (DA), ascorbic acid (AA), uric acid (UA), glucose (Glu), urea, Ca^2+^, and Fe^3+^. Each interferent with each NP was performed in triplicate way, reporting the error bars as standard deviation. The evaluation was made in a real sample following the methodology presented in Section 2.4, but instead using 200 µL of human serum obtained from blood by venipuncture in a vacutainer^(R)^ tube with anticoagulant from six healthy volunteers. To verify the calculated lactate concentration, an Accutrend^®^ Plus brand commercial lactometer (Roche, Basel, Switzerland) was used.

### 2.6. Characterization of γ-/α-Fe_2_O_3_ and γ-/α-Fe_2_O_3_/ZnO NPs

The crystal structure of the NPs was determined by an X-ray diffraction (XRD) measurement that was carried out by a Bruker X-ray diffractometer model PW3710 radiation CuK (alpha 1) (wavelength: 1.54 Å). X-ray photoelectron spectroscopy (XPS) analysis was performed using an instrument (Monochromatic Magics Thermo Scientifics, model K-Alpha+) in conditions of v = 1 scan/min, t = 20.5 s and CAE = 20. A L6S/L6 UV-Vis spectrophotometer was used to measure absorbance.

## 3. Results and Discussion

### 3.1. Characterization of γ-/α-Fe_2_O_3_ and γ-/α-Fe_2_O_3_/ZnO

From XRD analysis, the crystalline phase and size of the Fe_2_O_3_ and Fe_2_O_3_/ZnO nanoparticles were estimated. Analyzing the peaks belonging to the Fe_2_O_3_ (Figure 1), the diffraction peaks match the standard rhomboeric pattern, having the characteristics (012), (104), (110), (113), (024), (116), (214), (300), (119), and (220) planes of α-Fe_2_O_3_ (hematite), and the displaced peaks of planes (220), (311), (400), (422), and (511), which correspond to γ-Fe_2_O_3_ (maghemite), which suggests that a bi-phase γ-/α-Fe_2_O_3_ is formed. The result is consistent with the values already reported (JCPDS: 033-0664 and JCPDS: 04-0755). The formation of the bi-phase γ-/α-Fe_2_O_3_, according to other authors, can be be attributed to the synthesis and calcination method; also, the data processing was performed using a detailed analysis by DIFRAC EVA software (2.0) [22,23]. For Fe_2_O_3_/ZnO-based materials, the diffraction patterns attributed to α-Fe_2_O_3_ can be observed according to JCPDS: 033-0664, whereas other well-defined diffraction peaks corresponding to ZnO, assigned (100), (002), (101), (102), (110), (103), (112), and (201), were indicated, which correspond to JCPDS 36-1451 [24]. The Debye–Scherrer equation (D = κ λ/β cos θ) was used to estimate the size of the crystalline particles, where D is the crystal size in nm, λ is the X-ray wavelength (0.15406 nm), β is full-width at half-maximum of the peak, θ is the corresponding Bragg diffraction angle, and k is a constant (0.89) [25]. By consideration of the most representative peaks of each material, a particle size of 15 and 28.6 nm was calculated for γ-/α-Fe_2_O_3_ and γ-/α-Fe_2_O_3_/ZnO, respectively.

XPS analysis of γ-/α-Fe_2_O_3_ (Figure 2A) shows a first fine scan of Fe 2p, observing the binding energy of the electrons given in 2p^3/2^, 2p^1/2^ in the bands 711.45 eV and 734.41 eV, respectively [26]; the separation of the 2p doublet is 14.0 eV [27]. Specifically, the adjustment peaks that are located in the binding energies of 720.3, 724.2, 733.6, and 735.5 eV correspond to the tetrahedral Fe 2p^3/2^, octahedral Fe 2p^3/2^ structure, and the Fe 2p^1/2^ satellite structure, whereas in the energies 711.3, 713.2, 718.6, and 725.5 eV, there is a signal shift observed, which indicates a normal state of Fe^3+^ in the two samples [28]. In the analysis of γ-/α-Fe_2_O_3_/ZnO (Figure 2B), a second scan of Zn 2p was made, showing the peaks in the energy band 1022.1 and 1045.0 eV, which are attributed 2p^3/2^ and Zn 2p^1/2^, respectively, indicating that the oxidation state of Zn is 2^+^, whereas those located at 1021.4 and 1044.4 eV are attributed to Zn in the ZnO phase, as shown in the Appendix A [29]. By the analysis of the material based on γ-/α-Fe_2_O_3_/ZnO (Appendix A), it was found that the corresponding signals for γ-Fe_2_O_3_ and α-Fe_2_O_3_ at 532.5 eV can be attributed to a chemisorption process, whereas the peaks located at 560.5 and 584.5 eV are associated with the oxygen of the network in the framework of metal oxides (Zn/Fe) [27]. In addition, the atomic percentages of Zn, Fe, and O were calculated and adjusted to the fine-scan spectra Zn, Fe, and O, showing the Zn/Fe/O ratio of 1:2:4 in Appendix A.

The morphology of nanoparticles was characterized by S/TEM. According to Figure 3A, the γ-/α-Fe_2_O_3_ material showed an aggregation of the nanowire shape, with a length between 125 to 330 nm and a diameter of 18 nm (see Appendix A). These nanostructures have been previously obtained by the precipitation method and calcinated between 400 °C to 600 °C in air [30,31], or by hydrothermal methods [32], high pressures, or using other chemical precursors [33,34]. In this sense, several forms such as spindle, ellipsoid, spherical, and quasi-cubic [33,34] have been previously reported.

The architecture of the γ-/α-Fe_2_O_3_/ZnO material consisted of a mixture of nanowires and spherical-shape nanoparticles with an average size of 15 to 18 nm (Figure 3B). EDS analysis evidenced an aggregation of Fe, Zn, and O, as well as a random distribution between Fe_2_O_3_ with ZnO. Several morphologies of the Fe_2_O_3_/ZnO material also synthesized by the precipitation method have been reported, such as flowers [35], microflowers, nanosheets [24,36,37,38,39], hexagonal [40], spheres [41,42], core-shell nanoparticles [43,44], nanorods core-shell configuration [41], and nanotubes [42].

### 3.2. Evaluation of Peroxidase-like Activity of γ-/α-Fe_2_O_3_ and γ-/α-Fe_2_O_3_/ZnO

The evaluation of the synthesized nanoparticles, e.g., peroxidases-like activity, was carried out by a UV-vis absorbance technique at 655 nm, chosen from a wavelength scan between 400 to 800 nm (Appendix A). The TMB concentration (500 µM) was obtained from the slope in the plot of absorbance as a function of the TMB concentration in the presence of 1 mM H_2_O_2_ (Appendix A). The resulting graphs as a function of the H_2_O_2_ concentration for each material γ-/α-Fe_2_O_3_ (Figure 4A) and γ-/α-Fe_2_O_3_/ZnO (Figure 4B) showed a typical Michaelis–Menten behavior, whereas the maximum velocity (Vmax) and Michaelis–Menten constant (Km) were calculated from Lineweaver–Burk plots. Km values of 0.02 mM and 0.03 mM were estimated by using γ-/α-Fe_2_O_3_ and γ-/α-Fe_2_O_3_/ZnO, respectively. In both cases, the results indicate that the synthesized materials have high affinity for the substrate compared to other iron-based nanoparticles [21]. Values of 6.0 × 10^−8^ M s^−1^ and 8.9 × 10^−8^ M s^−1^ for Vmax were obtained for γ-/α-Fe_2_O_3_ and γ-/α-Fe_2_O_3_/ZnO, respectively, indicating a faster reaction rate, possibly due to a higher dispersion of the Fe_2_O_3_ catalyst with the intercalation within ZnO. These results are compared with other works already published (Table 1), where the lowest calculated Km was 0.013 mM (0.65 times lower than that obtained in this work) [43], and 2.25 × 10^−5^ M s^−1^ was the highest Vmax [19]. The results show that when Fe_2_O_3_ is doped with some material, the Vmax increases (see Table 1). The same behavior is observed when the ZnO material is incorporated, even in the bi-phase γ-/α-Fe_2_O_3_ compared to the already reported Fe_3_O_4_ [17].

**Table 1 biosensors-12-01025-t001:** Comparison of Km and Vmax values of γ-/α-Fe_2_O_3_ and γ-/α-Fe_2_O_3_/ZnO.

Catalyst	Km (mM)	Vmax (M s^−1^)	Ref.
γ-/α-Fe_2_O_3_	0.02	6.0 × 10^−8^	This work
γ-/α-Fe_2_O_3_/ZnO	0.03	8.9 × 10^−8^	This work
Fe_3_O_4_@C	0.014	13.35 × 10^−8^	[43]
Fe_3_O_4_	0.013	2.95 × 10^−8^	[17]
H_2_TCPP-γ-Fe_2_O_3_	0.013	21.14 × 10^−9^	[17]
GO-Fe_2_O_3_	0.71	5.31 × 10^−8^	[21]
γ-Fe_2_O_3_-SiO_2_	0.63	2.25 × 10^−5^	[19]
PB-Fe_2_O_3_	91.54	8.31 × 10^−8^	[44]
ZnFe_2_O_4_	1.66	7.74 × 10^−8^	[18]

From the H_2_O_2_ calibration curve, a linear range between 50–1000 µM (R^2^ 0.9956) was obtained using both nanoparticles, as well as 4.3 and 9.4 µM limits of detection (LOD) for γ-/α-Fe_2_O_3_ and γ-/α-Fe_2_O_3_/ZnO, respectively. The limits of quantification (LOQ) were estimated in 14.5 and 31.5 µM, respectively. This result is higher than those previously reported by other authors using iron oxides [16,20]. Likewise, colorimetric Au-Ag-based lactate sensors have reported LODs of 0.3 and 3 µM, whereas linear ranges are between 0.1–1000 µM, 0.8–90, and 90–500 µM. The results here obtained are comparable and, in some cases, higher, but with the advantage of using economically accessible materials [45]. The good activity, in terms of Km and Vmax values, obtained when γ-/α-Fe_2_O_3_ material was used could be attributed to the combination of both crystalline phases, which prevents the agglomeration and maximizes the reaction area.

### 3.3. Lactate Detection Using LOx, γ-/α-Fe_2_O_3_, and γ-/α-Fe_2_O_3_/ZnO

The nanoparticles of γ-/α-Fe_2_O_3_ and γ-/α-Fe_2_O_3_/ZnO combined with LOx were tested as a lactate sensor. For this procedure, it was assumed that the lactate oxidation reaction by LOx produces H_2_O_2_, which reacts with the TMB in the presence of the synthesized material, resulting in a color change that can be followed by UV-Vis spectroscopy. As can be seen in the upper part of Figure 5, a higher lactate concentration corresponds to a greater color intensity. The linear range for the lactate concentration was 50 to 1000 μM (R^2^ 0.9565) using both nanoparticles. The Km values estimated from the Lineweaver–Burk equation were 1.23 mM and 13.9 mM for γ-/α-Fe_2_O_3_ (Figure 5A) and γ-/α-Fe_2_O_3_/ZnO (Figure 5B), respectively. The LOD value was 50 μM for both nanomaterials. Already-published NPs have reported 0.0033 µM LOD and two linear ranges of 0.1–22 µM and 22–220 µM using noble metals, such as Au-Ag/C NC/LOx [45]. In the case of Ag/C, 0.35 µM LOD with two linear ranges of 1–30 µM and 30–900 µM was obtained [46]. Using Pt NPs, 0.6 µM LOD and a linear range of 2.5–100 µM were reported [47]. Hyun Jung Kim et al. reported 1 mM LOD and R^2^ 0.9684 using a paper-based colorimetric lactate sensor [48]. Though the LODs values are less than those obtained in this work, the range of detection is higher without the use of precious metals.

Furthermore, in order to evaluate the selectivity of the lactate sensor, γ-/α-Fe_2_O_3_ and γ-/α-Fe_2_O_3_/ZnO NPs were evaluated in the presence of different interferents. As can be seen in Figure 6, the absorbance was measured at 1 mM lactate in the presence of 0.5 mM of molecules that could interfere in the lactate quantification in biological fluids, such as dopamine (DA), acid ascorbic (AA), uric acid (UA), glucose (Glu), urea, and cations, such as Ca^2+^ and Fe^3+^ [49]. Using both Nps did not influence the lactate detection due to the Lox specificity and selectivity; however, by using LOx/γ-/α-Fe_2_O_3_/ZnO, the standard deviation values were lower. This result could be attributed to ZnO, which acts as a quenched agent for Fe^+3^ and for organic molecules such as dopamine [50,51].

After evaluating the catalytic activity of the NPs in the presence of lactate and their selectivity in the presence of some interferents, the characterization continued by using real serum samples. For this evaluation, blood serum samples were obtained from six volunteers, whose lactate concentration was measured using a blood serum meter, glucose/lactate Accutrend^®^ Plus brand. Subsequently, applying the same method previously used in lactate detection, blood serum (200 µL) was measured at 653 nm after the incubation time. For the estimation of lactate concentration in serum samples, the equation of the straight line of the calibration curve was used using both NPs presented in Figure 4. The obtained values are shown in Table 2, where the results are compared with those obtained by using a commercial lactometer. The relative error values are greater when LOx/γ-/α-Fe_2_O_3_ was used compared to those values obtained with LOx/γ-/α-Fe_2_O_3_/ZnO. Finally, the obtained values of lactate concentration obtained by using the NPs synthesized in this work are comparable with those previously obtained by other authors in body fluids such as sweat [52,53] and blood [49,54,55]. A better fit in the linearity of the lactate sensor is important because it reduces measurement errors and is easier to calibrate. The reported linear range for lactate biosensors is suitable for the concentrations found in different biological fluids. For example, in blood, it is between 0.5–2.2 mM and reaches up to 15 mM in conditions of prolonged exercise or metabolic pathologies [56]. In tears, it can be found between 1–5 mM and 16–30 mM in sweat, reaching up to 120 mM under exhaustive exercise conditions [57,58].

Although the tendency of actual sensors is the detection of lactate in sweat [59,60], colorimetric methods continue to be the simplest and most widely used. The results obtained here are promising due to more economical catalytic materials being used compared to those previously reported [45]. Furthermore, according to the difference of the isoelectronic point between ZnO and LOx of ~9.5 and 4.3, respectively [61], the presence of ZnO in the γ-/α-Fe_2_O_3_/ZnO structure could facilitate the LOx immobilization, whereas the Fe_2_O_3_-based nanomaterial allows the catalysis of H_2_O_2_, which means a synergetic contribution for the lactate colorimetric sensing. In this same sense, both nanoparticles evaluated are highly selective in the quantification of lactate in the presence of different biological interferents.

## 4. Conclusions

In this work, bi-phase γ-/α-Fe_2_O_3_ and γ-/α-Fe_2_O_3_/ZnO nanostructured materials were tested, e.g., peroxidase-like activity and lactate colorimetric sensors. The synthesis method was simple and economically accessible. Several physicochemical techniques, XRD, XPS and S/TEM, corroborated the structure and composition of the bi-phase nanoparticles. The proposed methodology in the presence of H_2_O_2_ indicated that γ-/α-Fe_2_O_3_ and γ-/α-Fe_2_O_3_/ZnO exhibit intrinsic peroxidase-like activity, whereas the kinetics and catalytic parameters evaluated by the Michaelis–Menten theory showed similar and superior values compared to those previously obtained with other iron oxide-based materials. The material based-on γ-/α-Fe_2_O_3_ presented a greater catalytic activity compared to other already-published peroxidase-like materials, whereas the γ-/α-Fe_2_O_3_/ZnO material retained its catalytic activity and exhibited a more stable signal. By a combination of synergetic properties of both metallic oxides (Fe_2_O_3_ and ZnO) using lactate oxidase enzyme, a cheap and simple lactate colorimetric sensor was constructed and evaluated using human serum samples with an acceptable quantification of lactate concentration compared to the values obtained with a commercial lactometer.

## Data Availability

Not applicable.

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
