# Peer review of "Comparative Colorimetric Sensor Based on Bi-Phase γ-/α-Fe2O3 and γ-/α-Fe2O3/ZnO Nanoparticles for Lactate Detection"

_biosensors, 2022, doi:10.3390/bios12111025_

Round 1
Reviewer 1 Report
The author developed bi-phase γ-/α-Fe2O3 and γ-/α-Fe2O3/ZnO nanostructured materials for lactate colorimetric sensors. The performance of materials was well characterized by XRD, XPS, and S/TEM, showing good applicability for lactometer development.
Minor comments
1. Author mentioned the developed materials have high affinity and good sensing performances compared to other iron-based nanoparticles such as GO-Fe2O3. However, I couldn't find differences in the sensing performance of developed materials to others, especially in table 1. Please describe the differences and advantages of developed materials compared to the other methods more clearly.
2. How did the author evaluate the LOD from H2O2 and Laccate calibration curves? what is the LOD of other nanomaterials for lactate sensing?
3. Please describe how the authors determined the linear region from calibration curves. is it based on the R2 linearity?
4. For a better understanding to broad readers, please briefly explain why linearity is important and what detection range is needed for lactate sensors.
Author Response
Reviewer #1
The author developed bi-phase γ-/α-Fe2O3 and γ-/α-Fe2O3/ZnO nanostructured materials for lactate colorimetric sensors. The performance of materials was well characterized by XRD, XPS, and S/TEM, showing good applicability for lactometer development.
Minor comments
- Author mentioned the developed materials have high affinity and good sensing performances compared to other iron-based nanoparticles such as GO-Fe2O3. However, I couldn't find differences in the sensing performance of developed materials to others, especially in table 1. Please describe the differences and advantages of developed materials compared to the other methods more clearly.
Dear reviewer, we agree with your observation. Additional discussion about this issue has been now included in the manuscript in order clarify the performance of Fe oxides-based materials.
- How did the author evaluate the LOD from H2O2and Laccate calibration curves? what is the LOD of other nanomaterials for lactate sensing?
Thank for your observation. The estimation of LOD has now described in the methodology section; further the comparison with other already reported LOD values for H2O2 and lactate has now described in Results and Discussion section. All changes are highlighted in yellow.
- Please describe how the authors determined the linear region from calibration curves. is it based on the R2linearity?
The determination of the linear region is based on the calibration curves according to their adjustment coefficient of determination (R2) to the best fit. In order to clarify this result, the R2's have been now added in the manuscript.
- For a better understanding to broad readers, please briefly explain why linearity is important and what detection range is needed for lactate sensors.
Dear Reviewer thank you for your suggestion. Additional information about the linearity in lactate sensors has been now included in the text.
Dear Reviewers, thank you very much in advance for your kind consideration to our work.
Please see the attachment.

Reviewer 2 Report
This paper presents a sound and well developed sensing system for lactate that performs well in real biological fluid samples. I recommend it be published provided the authors make some corrections to the presentation to make it more legible and understandable:
- Should be 3,3,5,5-tetramethylbenzidine in the abstract (not 3,30,5,50)
- Paper needs a fair bit of grammar correction; incorrect words are used in many places as well
- Lines 63-65 shouldn’t be in the intro as they are concluding statements. Instead a hypothesis should be presented to end the introduction
- The Fe2O3/ZnO synthesis is not clear in the methods section, were the two solutions mixed together at some point?
- Line 97, the source/reference of the molar extinction coefficient should be given
- Since you use a commercially available lactometer to compare to your nanoparticle system, you should devote more discussion to comparing your new system to this commercial system. Why is your system potentially better? If it’s cheaper, by how much. Is it easier to use. Etc…
- Ref 36 formatting is incorrect
Author Response
Reviewer # 2
This paper presents a sound and well developed sensing system for lactate that performs well in real biological fluid samples. I recommend it be published provided the authors make some corrections to the presentation to make it more legible and understandable:
- Should be 3,3,5,5-tetramethylbenzidine in the abstract (not 3,30,5,50)
Dear Reviewer thank you for your observations and suggestions that enrich our work. This is a mistake that has been now corrected in the manuscript.
- Paper needs a fair bit of grammar correction; incorrect words are used in many places as well
The manuscript has been carefully reviewed and corrected in order to improve the quality of the text.
- Lines 63-65 shouldn’t be in the intro as they are concluding statements. Instead a hypothesis should be presented to end the introduction.
We agree with your suggestion, the text has been modified in the manuscript.
- The Fe2O3/ZnO synthesis is not clear in the methods section, were the two solutions mixed together at some point?
The methodology has been now re-written in order to clarify the synthesis of materials.
- Line 97, the source/reference of the molar extinction coefficient should be given
We agree with the Reviewer observation, thus the reference has been now given in the text.
- Since you use a commercially available lactometer to compare to your nanoparticle system, you should devote more discussion to comparing your new system to this commercial system. Why is your system potentially better? If it’s cheaper, by how much. Is it easier to use. Etc…
Dear Reviewer, in the first instance, the use of the lactometer has the purpose of validating the lactate concentration in the real fluid measured with the materials proposed in this work. The preliminary results showed that our colorimetric sensor has possibilities to be used but additional evaluations and optimization are needed to ensure if this sensor is potentially better than a commercial sensor.
- Ref 36 formatting is incorrect
The reference has been corrected.
Please see the attachment.

Reviewer 3 Report
In the present study authors have developed two nanoparticles (NPs -bi-phase γ-/α-Fe2O3 and γ-/α-Fe2O3/ZnO) for their application in a lactate colorimetric sensor. NPs were used to evaluate the catalytic activity of peroxidase with the substrate 3,3’,5,5’-tetramethylbenzidine (TMB), obtaining a linear range of 50 to 1000 μM for both NPs and 4.3 μM and 9.4 μM of limit of detection (LOD), respectively. A sample of human serum was also tested and the results were compared with a commercial lactometer.
The study is well planned and most of the conclusions are supported by the data. The concept of this work is great; however, the following minor points need to be revised before this work can be published in the Journal of Biosensors.
Minor comments:
1- Abstract: “3,30,5,50-tetramethylbenzidine” should be corrected to “3,3′,5,5′-Tetramethylbenzidine”.
2- Scheme 1, It should be more informative with higher resolution of images and chemical names etc. thus the difference between TMB and TMB dimer is not clear based on the chemical structure in this figure.
3- In the introduction section other methods were introduced for the detection of lactate, “Lactate concentration is usually determined by HPLC, fluorometry, colorimetric tests [2], chemiluminescence, magnetic resonance spectroscopy, or electrochemical methods [3]. One of the biggest problem related with colorimetric methods is the poor sensibility at low concentrations of lactate; however is a method widely used due to simplicity, non-complex equipment and economically accessible [4,5].”, however the LOD and sensitivity of the current existing methods has not been compared with this work.
4- Page 5, “These nanostructures have been previously obtained by the precipitation method and calcinated between 400 °C to 600 °C in air [30,31], or by hydrothermal methods [32], high pressures or using other chemical precursors [33,34]. In this sense, several forms like spindle, ellipsoid, spherical, quasi-cubic [34,35,36,37] have been previously reported.”
Has the synthesis of NPs in this work been improved? Or it was identical to previously reported?
Author Response
Reviewer # 3
In the present study authors have developed two nanoparticles (NPs -bi-phase γ-/α-Fe2O3 and γ-/α-Fe2O3/ZnO) for their application in a lactate colorimetric sensor. NPs were used to evaluate the catalytic activity of peroxidase with the substrate 3,3’,5,5’-tetramethylbenzidine (TMB), obtaining a linear range of 50 to 1000 μM for both NPs and 4.3 μM and 9.4 μM of limit of detection (LOD), respectively. A sample of human serum was also tested and the results were compared with a commercial lactometer.
The study is well planned and most of the conclusions are supported by the data. The concept of this work is great; however, the following minor points need to be revised before this work can be published in the Journal of Biosensors.
Minor comments:
- Abstract: “3,30,5,50-tetramethylbenzidine” should be corrected to “3,3′,5,5′-Tetramethylbenzidine”.
Dear Reviewer thank you for your observations and suggestions that enrich our work. This is a mistake that has been now corrected in the manuscript.
- Scheme 1, It should be more informative with higher resolution of images and chemical names etc. thus the difference between TMB and TMB dimer is not clear based on the chemical structure in this figure.
We agree with your observation, then Scheme 1 has been modified as Reviewer suggestion.
- In the introduction section other methods were introduced for the detection of lactate, “Lactate concentration is usually determined by HPLC, fluorometry, colorimetric tests [2], chemiluminescence, magnetic resonance spectroscopy, or electrochemical methods [3]. One of the biggest problem related with colorimetric methods is the poor sensibility at low concentrations of lactate; however is a method widely used due to simplicity, non-complex equipment and economically accessible [4,5].”, however the LOD and sensitivity of the current existing methods has not been compared with this work.
Thank you very much for your observation. The information in the introduction section has been re-written in order to avoid confusions.
- Page 5, “These nanostructures have been previously obtained by the precipitation method and calcinated between 400 °C to 600 °C in air [30,31], or by hydrothermal methods [32], highpressures or using other chemical precursors [33,34]. In this sense, several forms like spindle, ellipsoid, spherical, quasi-cubic [34,35,36,37] have been previously reported.” Has the synthesis of NPs in this work been improved? Or it was identical to previously reported?
Dear Reviewer, the methodology proposed in this work for the synthesis of NPs is similar to already reported by other authors, just with slight changes, as example the control in the time of reagents addition, the control of temperature and the constant sonication facilitate the dispersion and lack of nanoparticles agglomeration. This information has been now included in the manuscript.
All cited references can be improved in the manuscript.
Dear Reviewer the references have been updated and improved.
The description of methods can be improved.
We agree with your suggestion. The methodology has been now re-written in order to clarify this issue.
The conclusions can be improved and an extensive editing of English language and style required.
The manuscript has been carefully reviewed and corrected in order to improve the quality of the text.
Dear Reviewers, thank you very much in advance for your kind consideration to our work.